# Fatigue Life Evaluation of Orthotropic Steel Deck of Steel Bridges Using Experimental and Numerical Methods

**Yong Zeng** [1,2,*] 🔾, **Shenxu Wang** [1,2], **Xiaofang Xue** [1,2], **Hongmei Tan** [1,2] 🔾 and **Jianting Zhou** [1,2] 🔾

1 State Key Laboratory of Mountain Bridge and Tunnel Engineering, Chongqing Jiaotong University, Chongqing 400074, China
2 Mountain Bridge and Materials Engineering Research Center of Ministry of Education, Chongqing Jiaotong University, Chongqing 400074, China
* Correspondence: yongzeng@cqjtu.edu.cn; Tel.: +86-13883238757

**Abstract:** Orthotropic steel deck (OSD) structures are widely used in the bridge deck system of rail transit bridges. Reducing the amplitude of the stress intensity factor is the most effective method to improve the fatigue life of OSD structures. In order to explore the fatigue crack propagation of the OSD structure and the factors affecting the amplitude of the structural stress intensity factor, linear elastic fracture mechanics and Paris' law is used for theoretical support in this paper. Firstly, a cable-stayed bridge of urban rail transit is taken as the research object, a full-scale segment model of the OSD structure is designed and static and fatigue tests are carried out. Based on the test data, the fatigue life of the structure is simulated and predicted. Finally, ABAQUS and Franc3D are used to analyze the influence of parameters, such as U-rib thickness, roof thickness and diaphragm thickness, of the OSD structure on the amplitude of the stress intensity factor. The test and FEM analysis results show that the thickness of diaphragm and the height of the U-rib have little effect on the fatigue life of the OSD structure, appropriately increasing the thickness of the top plate and U-rib has a positive significance for prolonging the fatigue life of the structure. In addition, it is also of reference value to the application of sustainability and the science of sustainable development.

**Keywords:** urban rail transit; orthotropic steel deck; fatigue testing; fatigue residual life; fatigue cracking; finite element method (FEM)

## 1. Introduction

Nowadays, environmental protection and energy saving have become really important, and urban rail transit has the advantages of being efficient, convenient, green and has the ability to relieve traffic congestion, etc. It has become one of the preferred modes of transport for most people. The deck plates of such structures are subjected to complex forces and are susceptible to fatigue cracking damage under high frequency fatigue cyclic loading. Moreover, the fatigue cracking of steel bridge deck has the characteristics of widespread, early and frequent occurrence. Once it occurs, the structure will be damaged before reaching the service life, resulting in huge economic losses. Therefore, it is necessary to discuss and study the fatigue cracking of the orthotropic steel bridge deck of urban rail transit bridges, and improve the bearing capacity of the structure and the safety performance of use.

Wu et al. [1] used a road rail dual-purpose arch bridge as a research object, determined the fatigue load spectrum parameters of the light rail support based on the actual light rail traffic, calculated the internal force history, and determined the constant fatigue load amplitude required for the fatigue test of three million cycles by using the linear cumulative damage criterion. Baietto et al. [2] proposed a method to predict crack expansion by cross-validating the measured test values with the calculated values from finite element software analysis. Baietto et al. [3] conducted fatigue tests on two sets of RD joint specimens with 15% and 75% weld penetration in order to research the fatigue resistance around

the rib plate (RD) weld in orthotropic steel bridge deck. Kainuma et al. [4] monitored the structural health of the Manhattan Bridge in real time based on FBG sensors, using deterministic and probabilistic methods to predict the remaining fatigue life of typical crack details. These studies are focused on the fatigue performance of the orthotropic bridge deck of highway or railway steel bridges, while there is less research on the orthotropic bridge deck of urban rail steel bridges, especially on the uniqueness of the urban rail steel bridge structure and load.

In order to better understand the fatigue problems of orthotropic anisotropic plate structures for urban rail transit, a rail transit cable-stayed bridge is used as the background to study the fatigue design and cracking problems of this structure. Relying on the design model dimensions of an orthotropic anisotropic steel bridge deck structure for an urban rail transit cable-stayed bridge, fatigue tests are carried out. Based on the fatigue cracking conditions of the tests, the finite element expansion simulations of the cracks are carried out, and the remaining fatigue life of the structure is predicted based on the simulation results and compared with the results of the tests. In the simulation of the crack test, the method combining ABAQUS and Franc3D is accurate and effective, which is conducive to the study of fatigue crack parameterization. The influence degree of the initial crack size, U-rib thickness, U-rib height, top plate thickness and cross partition thickness on the magnitude of the stress intensity factor at the leading edge of the cracks and the remaining fatigue life of the structure are investigated.

## 2. Fatigue Crack Expansion and Analysis of Calculation Results

### 2.1. Calculation of the Stress Intensity Factor

The cross-integration method (M-integration) is used to calculate the stress intensity factors. The cross-integration method enables the calculation of stress intensity factors for isotropic materials and generally anisotropic materials with Type I, II and III cracks ($K_I$, $K_{II}$ and $K_{III}$). The calculation principle of interactive integration is as follows.

Based on the fracture mechanics, it can be integrated as

$$J = \lim_{\Gamma_s \to 0} \int_{\Gamma_s} (w\delta_{ij} - \sigma_{ij}u_{ij})n_j \mathrm{d}\Gamma \tag{1}$$

where $n_j$ is the vector of the outer normal, $\Gamma_s$ is the crack tip perimeter, and $\Gamma_s$ is shown in Figure 1.

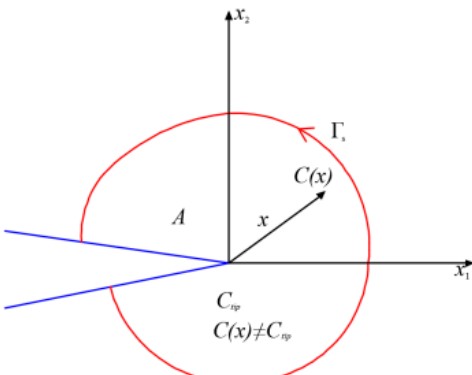

**Figure 1.** Integral contour.

$w$ is expressed as the strain energy density with the following expression [5,6]:

$$w = \frac{1}{2}\sigma_{ij}\varepsilon_{ij}^m \tag{2}$$

where $\varepsilon_{ij}^m$ indicates partial strain.

$$J = \int_A (\sigma_{ij}\frac{\partial u_i}{\partial x_1} - w\delta_{1j})\frac{\partial q}{\partial x_j}\mathrm{d}A + \int_A (\sigma_{ij}\frac{\partial u_i}{\partial x_1} - w\delta_{1j})q\mathrm{d}A \tag{3}$$

where $q$ indicates the crack expansion vector.

Superimposing the real stress field around the crack with the auxiliary fields ($u^{aux}$, $\sigma^{aux}$, $\varepsilon^{aux}$) on each other to obtain [7–9]

$$\begin{aligned} J^S &= \int_A \left\{ (\sigma_{ij} + \sigma_{ij}^{aux})(\frac{\partial u_i}{x_1} + \frac{\partial u_i^{aux}}{x_1}) - \frac{1}{2}(\sigma_{ik} + \sigma_{ik}^{aux})(\varepsilon_{ik} + \varepsilon_{ik}^{aux})\delta_{1j} \right\}\frac{\partial q}{\partial x_1}\mathrm{d}A \\ &+ \int_A \left\{ (\sigma_{ij} + \sigma_{ij}^{aux})(\frac{\partial u_i}{x_1} + \frac{\partial u_i^{aux}}{x_1}) - \frac{1}{2}(\sigma_{ik} + \sigma_{ik}^{aux})(\varepsilon_{ik} + \varepsilon_{ik}^{aux})\delta_{1j} \right\}q\mathrm{d}A \end{aligned} \tag{4}$$

$M$ can be rewritten as the interaction credit

$$\begin{aligned} M &= \int_A \left\{ \sigma_{ij}u_{i,1}^{aux} + u_{i,1}\sigma_{ij}^{aux} - \frac{1}{2}(\sigma_{ik}\varepsilon_{ik}^{aux} + \sigma_{ik}^{aux}\varepsilon_{ik}^m)\delta_{1j} \right\}q_{,j}\mathrm{d}A \\ &+ \int_A \left\{ \sigma_{ij}u_{i,1}^{aux} + u_{i,1}\sigma_{ij}^{aux} - \frac{1}{2}(\sigma_{ik}\varepsilon_{ik}^{aux} + \sigma_{ik}^{aux}\varepsilon_{ik}^m)\delta_{1j} \right\}q\mathrm{d}A \end{aligned} \tag{5}$$

Substituting the split-tip Auxiliary Field A into the above equation, then [10–12]

$$\begin{aligned} M &= \int_A \left\{ \sigma_{ij}u_{i,1}^{aux} + u_{i,1}\sigma_{ij}^{aux} - \sigma_{ik}\varepsilon_{ik}^{aux} \right\}q_{,j}\mathrm{d}A \\ &+ \int_A \left\{ \sigma_{ij}u_{i,1}^{aux} + u_{i,1}\sigma_{ij}^{aux} - \sigma_{ik}\varepsilon_{ik}^{aux} \right\}q\mathrm{d}A \end{aligned} \tag{6}$$

According to Equations (1) and (3), Equation (6) can be expressed as

$$J^S = J + J^{aux} + M \tag{7}$$

For open cracks, the relationship between the $M$ integral and the Type I stress, the intensity factor is:

$$\begin{aligned} M &= \frac{2}{E^*}K_\mathrm{I}K_\mathrm{I}^{aux} \\ K_\mathrm{I} &= \frac{E^*}{2}M \end{aligned} \tag{8}$$

where $K_\mathrm{I}^{aux} = 1$.

For plane strain state, $E^* = E/(1-v^2)$, where $v$ indicates the Possion ratio.

The normalized stress intensity factor ($K_\mathrm{I}$, $K_\mathrm{II}$ and $K_\mathrm{III}$) calculation results of the initial crack front are shown in Figure 2. The calculation at the position 0.5 corresponds to the calculated value of the stress intensity factor at the deepest point in the short semi−axis of the semi−elliptical crack, and SIF represents the stress intensity factor value.

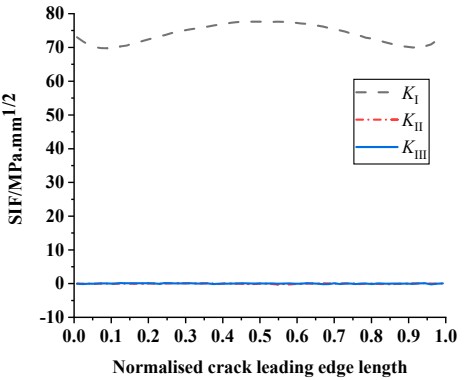

**Figure 2.** Normalized stress intensity factor at initial crack front.

As can be seen in Figure 2, the initial crack Type I stress intensity factor is symmetrically distributed, with larger values at the middle of the crack leading edge (short semi−axial crack tip) and at the ends of the long semi−axis, with a maximum value of up to 77.67 MPa·m$^{1/2}$, which shows that the initial crack expands relatively quickly at the ends of the long and short semi−axes; the stress intensity factors of Type II and Type III fluctuate around 0 MPa·m$^{1/2}$, which is much smaller than the stress intensity factors for Type II and Type III, which fluctuate around 0 MPa.mm and are much smaller than those for Type I. It shows that the crack is a composite crack dominated by Type I cracks.

### 2.2. Fatigue Life Assessment Method Based on Fracture Mechanics

In the assessment of the fatigue life of metallic structures, many studies have been conducted and a many achievements have been made. Based on the fatigue crack expansion of metallic materials, the mathematical model of the fatigue crack expansion, as shown in Figure 3, was obtained [13,14].

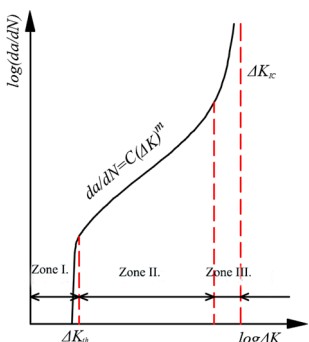

**Figure 3.** Fatigue crack expansion curves.

As can be seen in Figure 3, crack expansion can be divided into three stages. When the crack tip's stress intensity factor amplitude value, $\Delta K$, is less than the threshold value, $\Delta K_{th}$, the crack is in the first stage, and the crack does not expand. When the crack tip's $\Delta K$ is higher than $\Delta K_{th}$, the crack is in the second stage, and the crack will gradually and smoothly expand, which is currently described by the Paris law for this stage of crack expansion. When the crack tip's, $\Delta K$, is close to $\Delta K_{IC}$, the crack is in the third stage, and it expands rapidly and fractures unsteadily.

In this paper, the Paris law will be used to estimate the fatigue life of components; the Paris law is as follows [15]:

$$\frac{da}{dN} = C(\Delta K)^m \tag{9}$$

where $N$ is the number of stress cycles; $\Delta K = K_{\max} - K_{\min}$ is the amplitude value of the stress intensity factor; $K_{\max}$、$K_{\min}$ are the maximum and minimum values of $K$ under cyclic loading, respectively; $C$ and $m$ represent the material parameters associated with the test conditions.

Integrating Equation (9) over the crack length, $a$, an expression for the remaining fatigue life of the member can be obtained [16–18].

$$N = \int_{a_0}^{a_{cr}} \frac{1}{C(\Delta K)^m} da \tag{10}$$

where $a_0$ is the initial crack length, and $a_{cr}$ is the critical crack length.

When $m$ = 2, the expression for calculating the remaining life of a fatigue crack is

$$N = \frac{1}{C(\Delta K)^m} \ln(\frac{a_{cr}}{a_0}) \tag{11}$$

When $m \neq 2$, the expression for calculating the remaining life of a fatigue crack is

$$N = \frac{1}{C(\Delta K)^m (0.5m - 1)} \left( a_0^{1-0.5m} - a_{cr}^{1-0.5m} \right) \tag{12}$$

The Paris law is simple and has a small margin of error, and is one of the most commonly used formulas for predicting the remaining fatigue life of a steel component [19–21].

### 2.3. Simulation of Crack Extension

A three-dimensional crack growth is predicted in the following steps:

(1)  The local torsion angle can be calculated based on the stress at the leading edge of the local crack in the local co-ordinate system, shown in Figure 4, where the stress is determined by the local stress intensity factor.

(2)  Solve for the length of the local extension at each point.

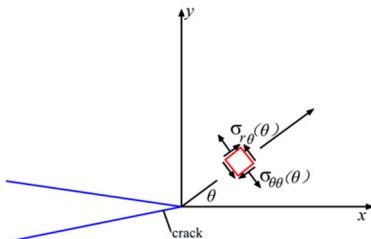

**Figure 4.** Local crack leading edge stress.

The leading edge of the new crack after extension is smoothed, and the leading edge of the crack is externally inserted outside the free surface of the structure.

A schematic diagram of the predicted crack front in Franc3D is shown in Figure 5.

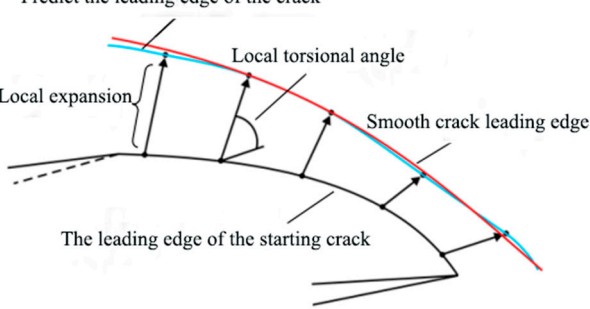

**Figure 5.** Local crack leading edge stress.

### 2.4. The Steps of Calculating Fracture Parameters by Finite Element Method

Fatigue crack growth and fracture parameters are conducted by the finite element method as follows [22–24]:

(1)  Building the finite element model

In finite element software (e.g., ABAQUS), the whole model without cracks is created. To increase the speed of calculation, the elements in the crack extension area are usually set as a group defined as a sub-model, and the non-crack extension area is defined as the key pattern;

(2)  Introduction of initial cracking

The shape and size of the initial crack are inserted into the sub-model according to the operation flow of the new defect wizard, and the mesh is regenerated.

(3)  Finite element calculation

The cracked sub-model is reassembled with the master pattern as a whole model containing the cracks, and it is automatically submitted to the finite element software for calculation;

(4)    Prediction of crack growth

The stress results and the stress intensity factors of each node at the crack front are read, the expansion step or the number of cyclic load actions are set, the position of the crack front are updated, and the mesh of the sub-model is remeshed;

(5)    Performing new finite element calculations

When the obtained results do not meet the user-defined stop conditions, the crack front position will continue to be updated and calculated. If the obtained results meet the user-defined stop conditions, the propagation analysis will be completed.

## 3. Model Design of Fatigue Test

### 3.1. Model Design

The test is based on an urban rail transit cable-stayed bridge with a main span of 340 m. The size of the model is designed according to the structural size of the steel deck of the bridge, and the model is simplified according to the actual situation. The Q345qD steel material is used as the model material. Three U-ribs with a spacing of 600 mm are set in the transverse bridge direction, and three diaphragm plates with a spacing of 827 mm are set in the longitudinal bridge direction. The 12-mm-thick steel plates are used at both ends for heads, and the size is 1800 mm × 121 mm × 281 mm. The steel box is placed at the lower part of both ends of the bridge deck and used as a support to keep the deck horizontal. The overall dimension of the model is 3000 mm × 1800 mm, and the thickness of the deck, diaphragm and U-shaped longitudinal rib is 16 mm, 12 mm and 8 mm, respectively. Since the restraints between the diaphragm and the U-rib will produce secondary bending stress, in order to reduce this stress, appropriate notches are usually made at the intersection between the lower part of the U-rib and the diaphragm, and the notch radius of the model designed for this fatigue test is 25 mm. The front view and three-dimensional view of the model are shown in Figures 6 and 7, respectively.

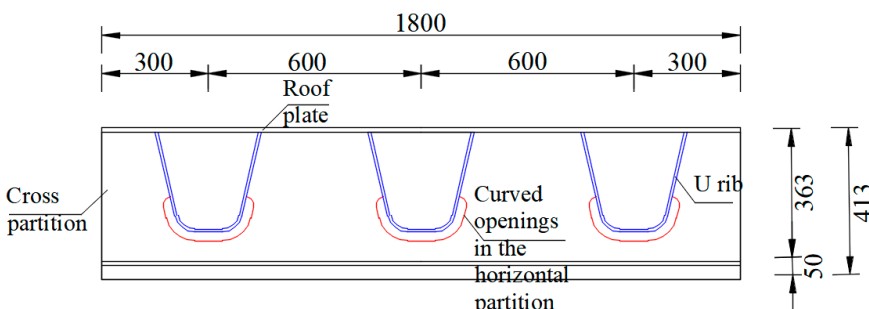

**Figure 6.** Front view of the full-scale model/mm.

### 3.2. Layout of Measuring Points

In order to investigate the stress and fatigue performance of the OSD of the test model, strain measuring points are arranged at the joint of longitudinal ribs and diaphragms, the joint of longitudinal ribs and top plates and the edges of diaphragm openings. The measuring point numbers of the middle diaphragm, side diaphragms, longitudinal ribs and top plates are ZHGB1-1 to ZHGB1-35, BHGB1-1 to BHGB1-19, BHGB2-1 to BHGB2-5, U1 to U6, and P1 to P6, respectively. The strain gauges used in this test were BX120-3CA, with a sensitive grid size of 3 mm × 2 mm and a resistance of 120 Ω ± 0.1%. A total of 71 strain gauges were laid out for the test, of which 35 were laid out in the middle cross partition, 19 in Side Cross Partition 1 (BHGB1), five in Side Cross Partition 2 (BHGB2), which was symmetrical to BHGB1, six in the U-rib, and six in the top plate. The specific location of

the measurement points is shown in Figure 8, where the black numbers in Figure 8a indicate the location of the intermediate measurement points, the red numbers in the brackets indicate the location of the Side Bulkhead 1 (BHGB1) measurement points, and the blue numbers indicate the location of the Side Bulkhead 2 (BHGB2) measurement points.

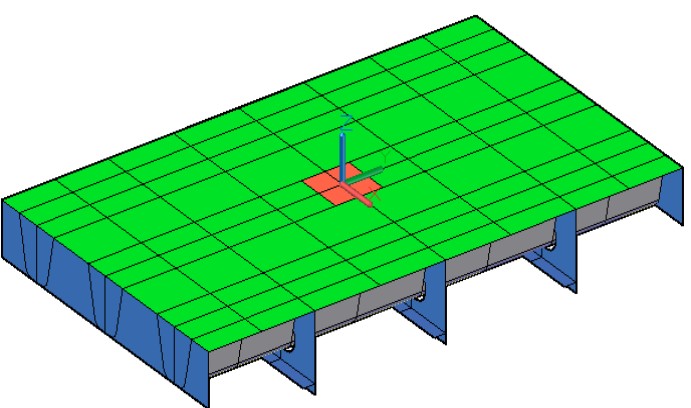

**Figure 7.** Three-dimensional view of the full-scale sectional test model.

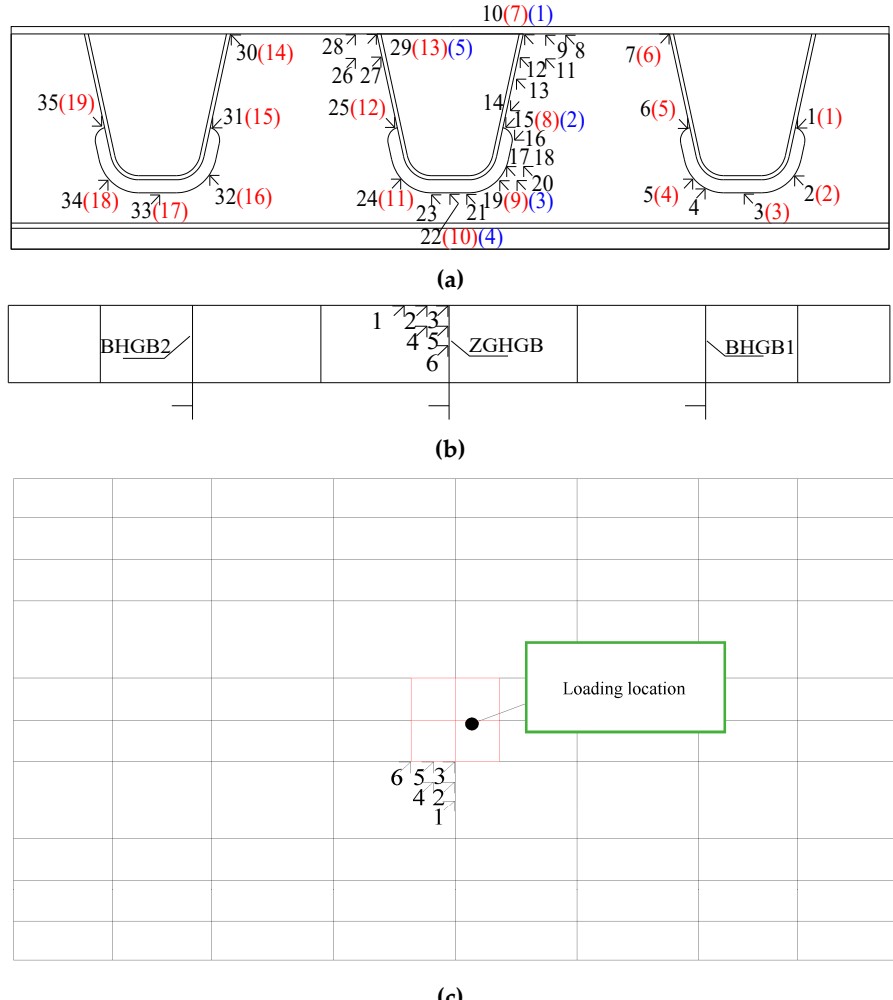

**Figure 8.** Layout of measuring points: (**a**) Layout of measuring points of the middle diaphragm; (**b**) Layout of U-rib measuring points; (**c**) Layout of roof measuring points.

## 4. Fatigue Test Results

### 4.1. Static Load Test Results

Before the static load test, the strain gauges are checked for adhesion and whether they are in good condition, after which at least three preloads are carried out and data are collected. The loading principle of the static load test is to load step by step from zero to the maximum static load value, and then all loads are removed symmetrically. After each load, a three-minute pause was taken to collect strain and displacement data. The maximum value of the static load for this test is 131.6 kN and the whole loading process is 0 kN, 10 kN, 30 kN, 70 kN, 110 kN, 131.6 kN, 110 kN, 70 kN, 30 kN, 10 kN and 0 kN, respectively. The above steps are operated at least twice, and the average values are calculated in order to reduce the dispersion of the test data.

The comparison results between the measured values and the calculated values of some key test points are listed in Table 1.

**Table 1.** Comparison of measured and calculated values at selected measurement points under static load.

| Location | Measurement Points | Measured Values/MPa | Calculated Values/MPa |
|---|---|---|---|
| | P2 | 56.6 | 55.5 |
| Roof plates | P3 | 156.6 | 160.1 |
| | P4 | 112.7 | 113.3 |
| | U2 | 53.4 | 55.0 |
| U-ribs | U3 | 64.7 | 63.9 |
| | U4 | 36.6 | 35.2 |
| | BHGB1-6 | 9.3 | 9.6 |
| | BHGB1-8 | 12.4 | 11.9 |
| Side dividers 1 | BHGB1-10 | 14.2 | 14.5 |
| | BHGB1-12 | 11.5 | 12.6 |
| | BHGB1-14 | 6.3 | 6.8 |
| | ZHGB1-7 | 29.0 | 29.3 |
| | ZHGB1-9 | 39.8 | 40.2 |
| | ZHGB 1-10 | 89.3 | 90.0 |
| | ZHGB 1-19 | 21.6 | 21.9 |
| Mid-transom bulkhead | ZHGB 1-22 | 50.9 | 51.2 |
| | ZHGB 1-31 | 19.3 | 18.7 |
| | ZHGB 1-33 | 29.1 | 30.3 |
| | ZHGB1-35 | 11.1 | 10.9 |

### 4.2. Fatigue Test Process

The fatigue load range for this test is from 10 kN to 131.6 kN. When the cyclic loading times reach 50,000 times, 250,000 times, 500,000 times, 750,000 times, 1,000,000 times, 1,250,000 times, 150,000 times, 750,000 times and 2,000,000 times the machine is stopped for a static load test, the strain values and displacement values are measured, and the model is checked [25–27].

If the test specimen is still in good condition when the number of loadings reaches two million, the load amplitude is increased and the fatigue test is continued. When the loading times reach 2.25 million times, 2.5 million times, 2.75 million times, 3.0 million times, 3.25 million times, and 3.5 million times the test is stopped and the strain, displacement values are measured and the condition of the model is checked. If the specimen has not been damaged after 3.5 million cycles of loading, the test is stopped. The fatigue load test process is shown in Figure 9.

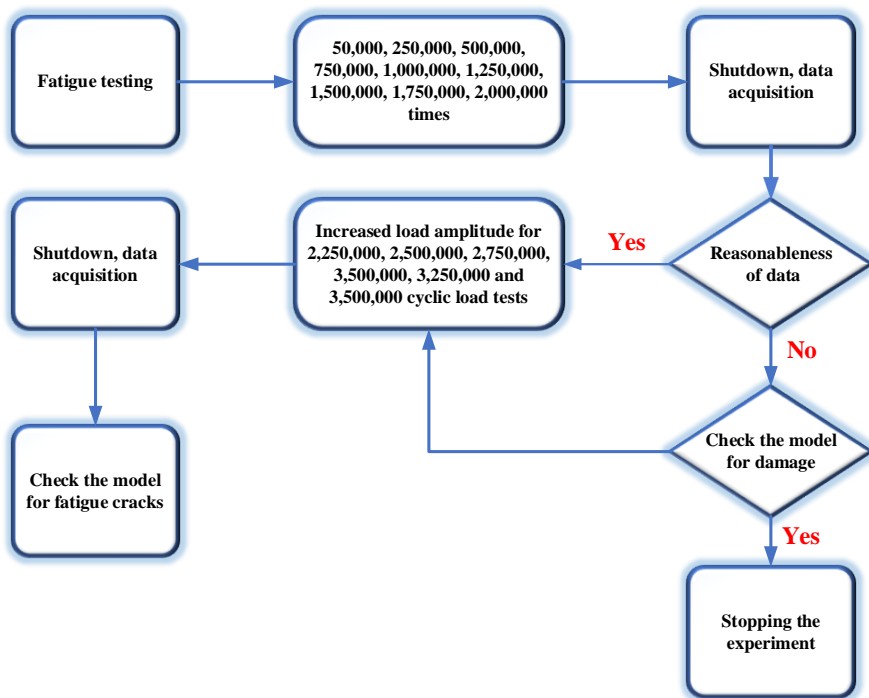

**Figure 9.** Flow chart of fatigue load test.

### 4.3. Fatigue Cracking's Location

At the completion of 2 million cycles of loading, the fatigue load amplitude was 121.6 kN, and no fatigue crack was found after careful observation. At 2 million to 2.5 million and 2.5 million to 3 million cycles, the fatigue load amplitude was increased to 1.25 and 1.5 times the original one, respectively, and no crack that was visible to the naked eye appeared on the test model. At 3 million to 3.25 million cycles, the fatigue load amplitude went up by 1.75 times the original one, and a crack of 15.1 cm in length appeared near the boundary of the loading location. At 3.25 million to 3.5 million cycles, a crack of 18.6 cm appeared. The fatigue cracks in the test model are shown in Figure 10.

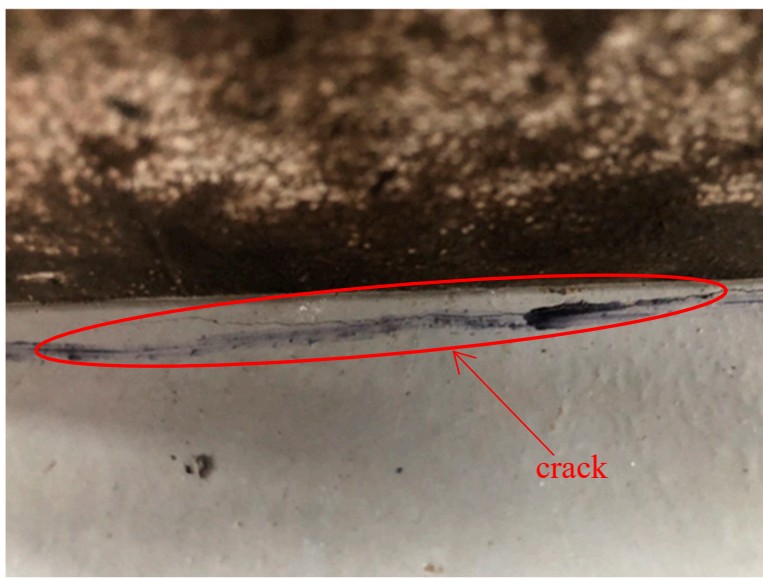

**Figure 10.** Schematic diagram of fatigue crack location.

## 5. Fatigue Crack Life Prediction

*5.1. Fatigue Cracking's Location*

From the test results, it can be seen that the cracks appear at the junction of the top plate–U-rib–middle diaphragm and crack downwards from the top surface of the top plate, so this section is devoted to an extended analysis of the cracks at this point. According to the study of fatigue cracking by Ya et al. [28] and Chen Chuanyao, Liu Yanping et al. [29], the shape of the fatigue crack can usually be approximated as a semi-ellipse, and it is assumed in this section that the initial half short axis of the crack is 0.1 mm and the half long axis is 0.15 mm; the structure is considered to be damaged when the depth of the crack penetrates the top plate, and the corresponding number of cyclic loads is the fatigue residual life of the OSD model structure.

ABAQUS modelling software is used to build a whole model without an initial crack, and the load and boundary conditions of the whole model are the same as the test model. The model and calculation results are shown in Figures 11–14. The elements in the crack extension area are set into a group and defined into a sub-model; then, the model is imported into Franc3D, so an initial crack is inserted into the sub-model and the mesh is regenerated [30,31].

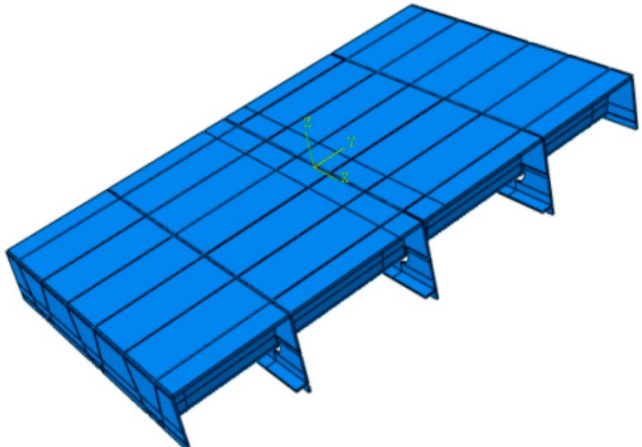

**Figure 11.** Geometric model.

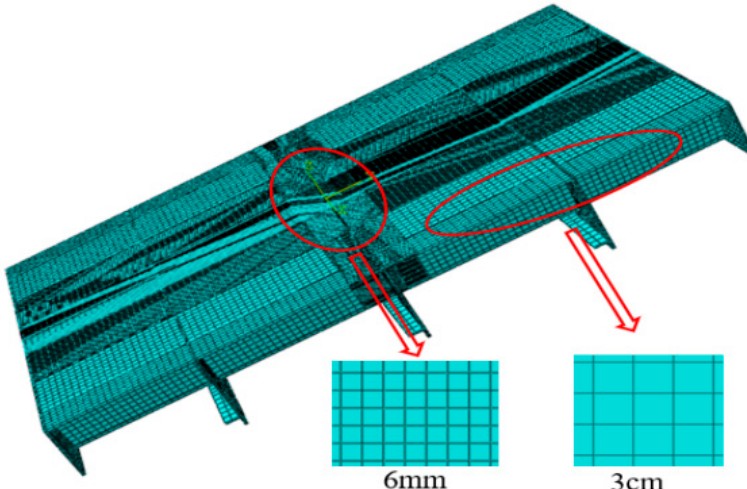

**Figure 12.** Finite element model.

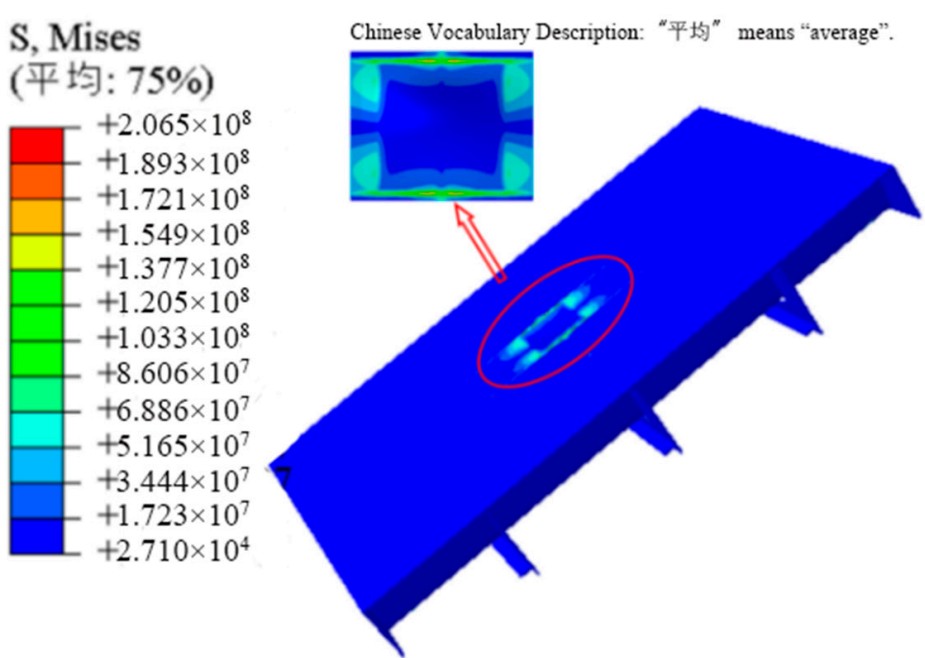

**Figure 13.** Finite element von Mises stress nephogram/Pa.

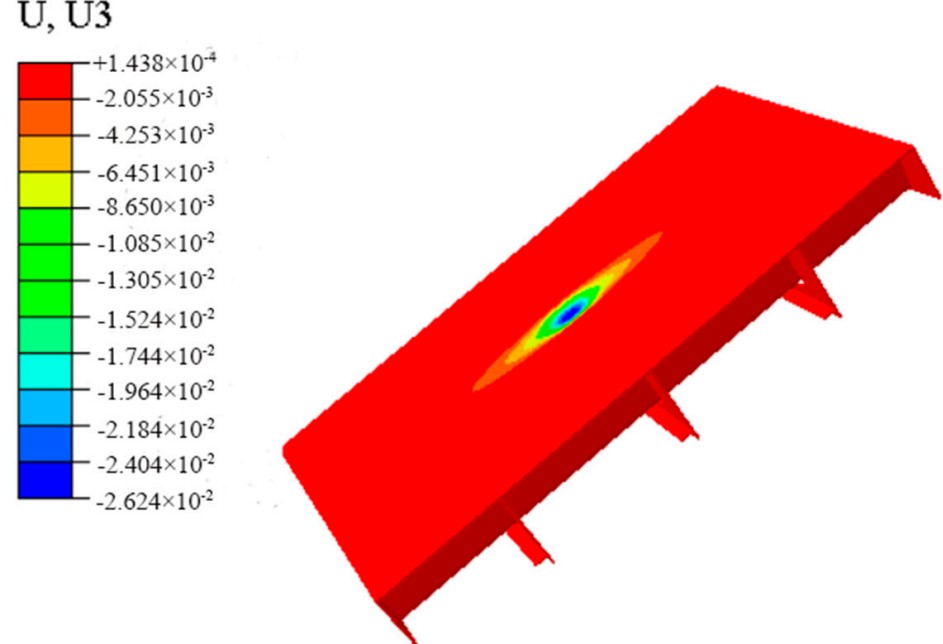

**Figure 14.** Finite element vertical displacement nephogram/mGeometric model.

At this time, the leading edge of the crack is divided into three rings of elements with a radius of one tenth of the short semi−axis of the crack. The initial crack front mesh division is shown in Figure 15. Finally, the sub-model and the key pattern are re-assembled into a model with the crack, and are submitted together to ABAQUS for finite element calculations.

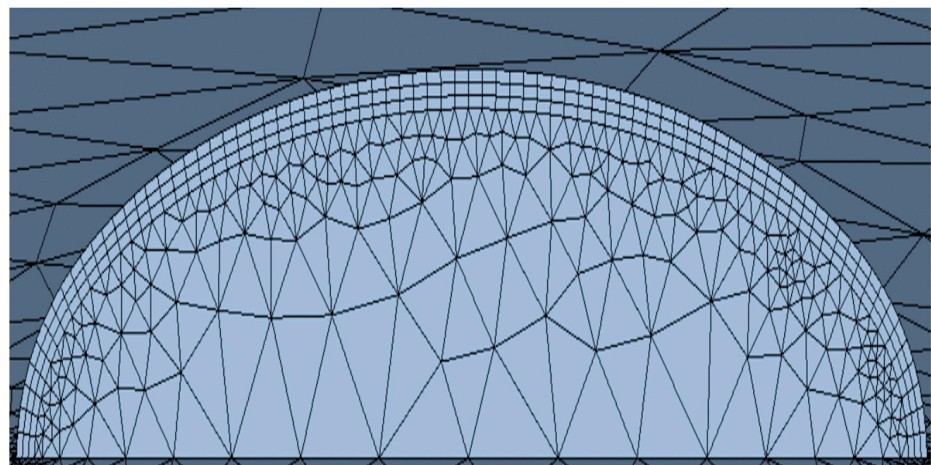

**Figure 15.** Meshing of the crack front of the model.

Typically, the extension step for each node on the leading edge of the crack is different. Two methods of calculating the extension step are provided in Franc3D software. One is to specify the extension step for the node located at the median value of the stress intensity factor; the extension step for all other nodes is obtained by appropriate scaling, and the other is to specify the number of cycles of the load and solve directly for the extension step for each node according to the Paris law [32–34]. In this section, the former method was used to calculate the expansion step for the nodes on the leading edge of the crack, specifying the expansion step at the median stress intensity factor to be less than or equal to fifteen percent of the characteristic size of the crack for a total of 37 expansion steps.

The specific extension step lengths for each step are shown in Table 2, and the crack changes throughout the extension process are shown in Figures 16–22. Due to the relatively large number of extension steps, only the crack changes for some key extension steps are shown [35,36].

**Table 2.** Crack extension sizes per step.

| Extension Steps | Current Size/mm | Extended Step Size/mm | Extension Steps | Current Size/mm | Extended Step Size/mm |
|---|---|---|---|---|---|
| 0 | 0.100 | 0.015 | 19 | 1.424 | 0.213 |
| 1 | 0.115 | 0.017 | 20 | 1.637 | 0.245 |
| 2 | 0.132 | 0.020 | 21 | 1.882 | 0.282 |
| 3 | 0.152 | 0.023 | 22 | 2.164 | 0.325 |
| 4 | 0.175 | 0.026 | 23 | 2.489 | 0.373 |
| 5 | 0.201 | 0.030 | 24 | 2.863 | 0.429 |
| 6 | 0.231 | 0.035 | 25 | 3.292 | 0.494 |
| 7 | 0.266 | 0.040 | 26 | 3.786 | 0.568 |
| 8 | 0.306 | 0.046 | 27 | 4.354 | 0.653 |
| 9 | 0.352 | 0.053 | 28 | 5.007 | 0.751 |
| 10 | 0.405 | 0.061 | 29 | 5.758 | 0.864 |
| 11 | 0.465 | 0.070 | 30 | 6.621 | 0.993 |
| 12 | 0.535 | 0.080 | 31 | 7.614 | 1.142 |
| 13 | 0.615 | 0.092 | 32 | 8.757 | 1.313 |
| 14 | 0.708 | 0.106 | 33 | 10.070 | 1.510 |
| 15 | 0.814 | 0.122 | 34 | 11.580 | 1.737 |
| 16 | 0.936 | 0.140 | 35 | 13.318 | 1.341 |
| 17 | 1.076 | 0.161 | 36 | 14.659 | 1.341 |
| 18 | 1.238 | 0.186 | 37 | 16.000 | / |

Figures 16–22 are shown below.

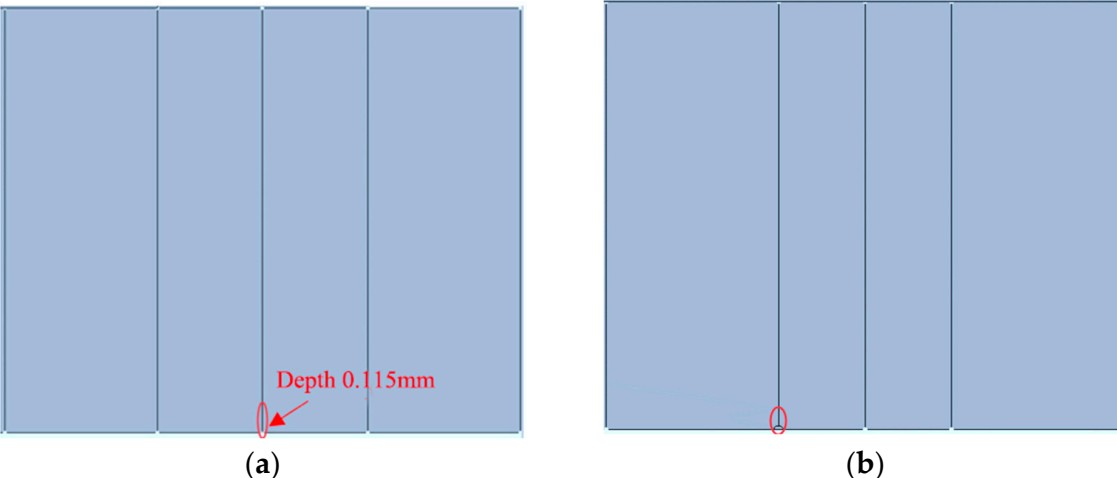

**Figure 16.** Crack shape simulation in Step 1: (**a**) Side view of the crack; (**b**) Front view of the crack.

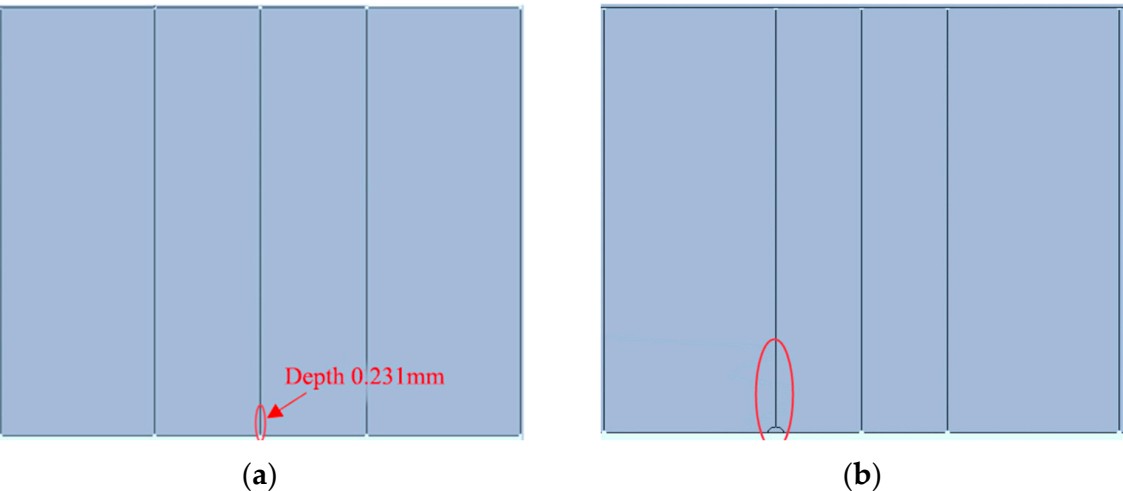

**Figure 17.** Crack shape simulation in Step 6: (**a**) Side view of the crack; (**b**) Front view of the crack.

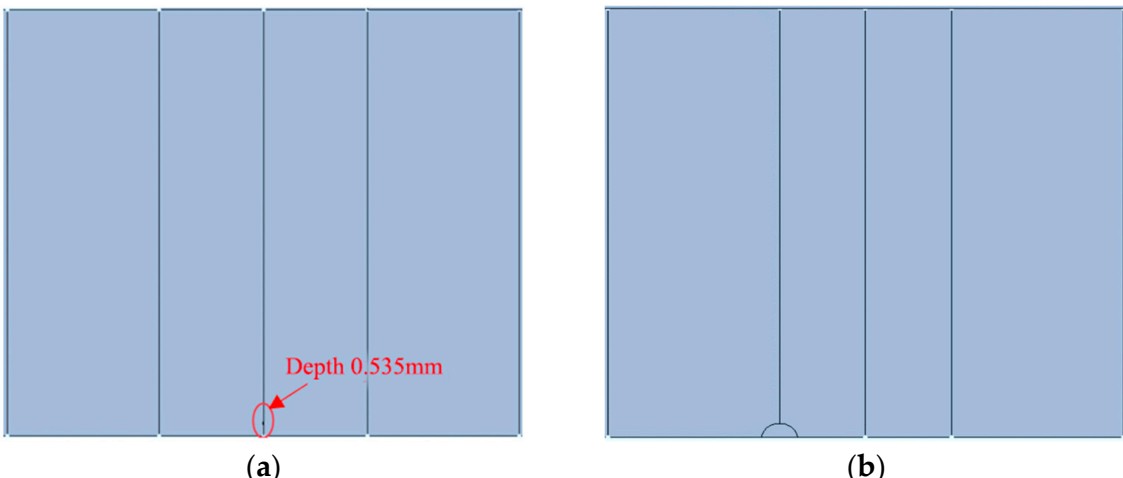

**Figure 18.** Crack shape simulation in Step 12: (**a**) Side view of the crack; (**b**) Front view of the crack.

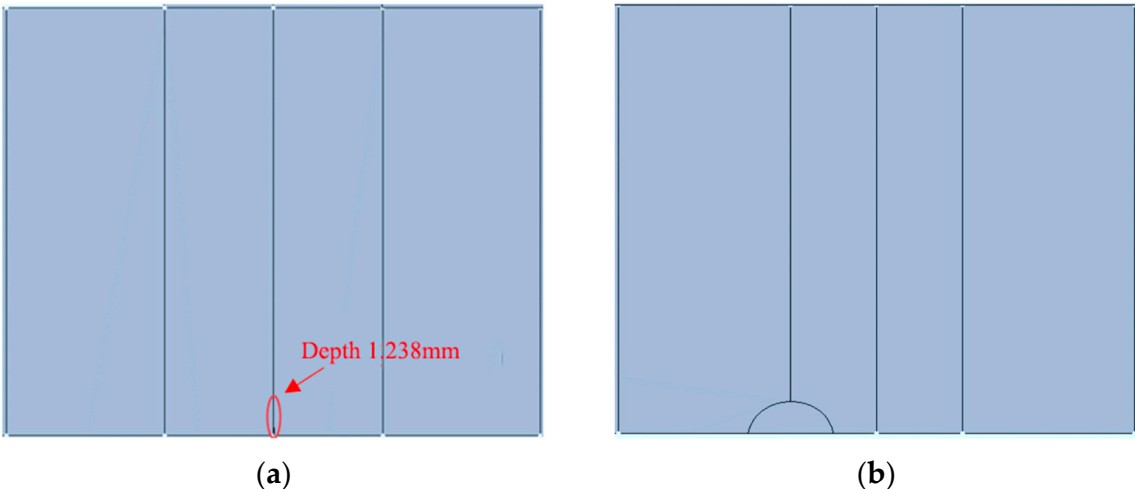

**Figure 19.** Crack shape simulation in Step 18: (**a**) Side view of the crack; (**b**) Front view of the crack.

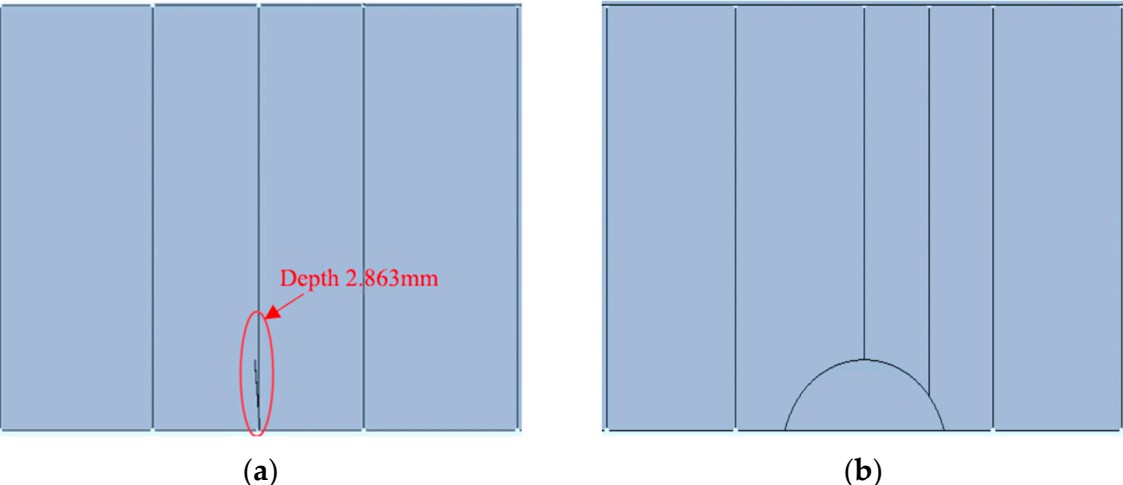

**Figure 20.** Crack shape simulation in Step 24: (**a**) Side view of the crack; (**b**) Front view of the crack.

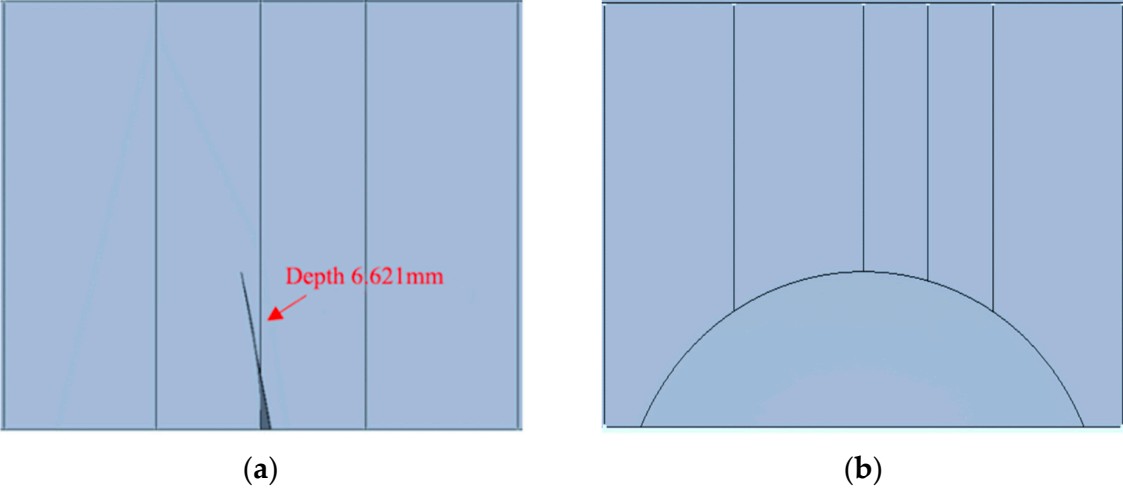

**Figure 21.** Crack shape simulation in Step 30: (**a**) Side view of the crack; (**b**) Front view of the crack.

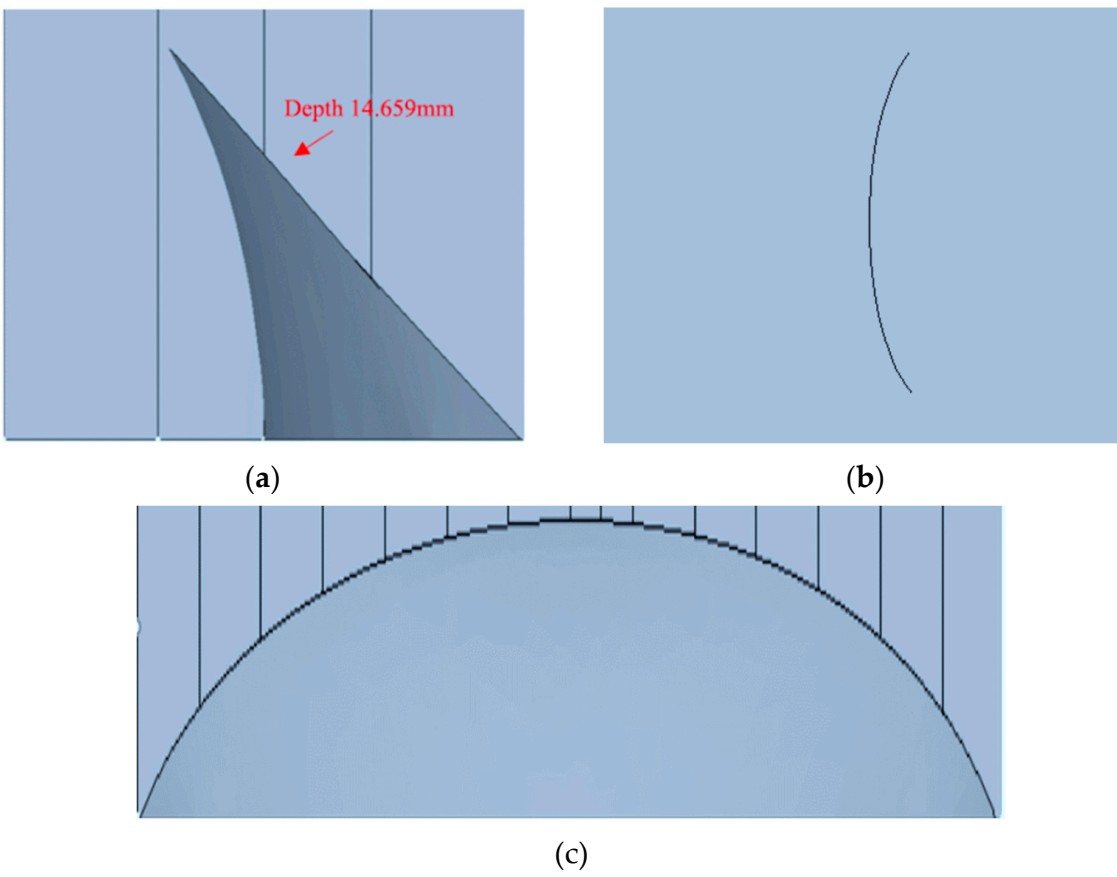

**Figure 22.** Crack shape simulation in Step 36: (**a**) Side view of the crack; (**b**) Top view of the crack; (**c**) Front view of the crack.

The stress intensity factor for each key extension step was calculated using the method of calculating the stress intensity factors for the initial crack, and the results are shown in Figure 23. Due to the large number of extension steps, only some of the extension steps are listed in Figure 23.

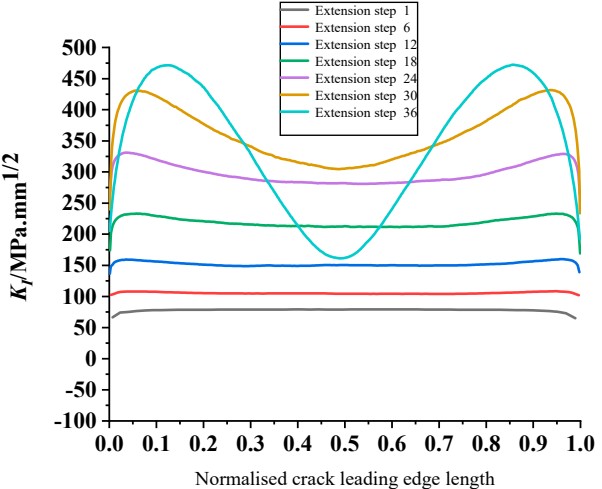

**Figure 23.** Calculation of stress intensity factors for the leading edge of some extended−step cracks.

According to Figure 23, the stress intensity factor at the middle point of the leading edge of the crack is larger than the other points when the crack is first expanded, so the expansion rate at this point is relatively large. With the increases of the expansion steps,

the stress intensity factors at about 5–10% from the two end points of the long half-axis are obviously larger than the stress intensity factors at other points, so the expansion rate of the crack along the long half-axis is larger than the expansion rate along the short half-axis. At the same time, as the crack expands, the stress intensity factor at the middle point of the leading edge of the crack shows a trend of firstly increasing and then decreasing. The reason for this phenomenon may be the fact that the crack expansion in the length direction is based on the expansion fitting in the depth direction, which makes the expansion rate in both directions slightly different, which in turn may lead to an increase in the ratio of the long semi axis to the short semi axis in the expansion process, and the gradual flattening of the crack shape, which is no longer stable, also causes a change in the stress intensity factor at the mid-point of the leading edge of the crack [37–39]. The observation of the model test shows that the cracks are indeed very flat and long semi-elliptical in shape, which shows that the expansion of the fatigue cracks is objective and inevitably related to the special stress pattern of the OSD structure. The variation in the length and direction of the crack's long and short semi-axes during the whole expansion process are shown in Figure 24.

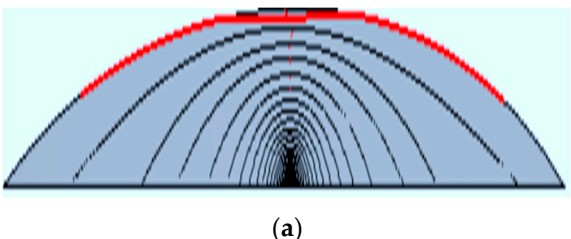
**(a)**

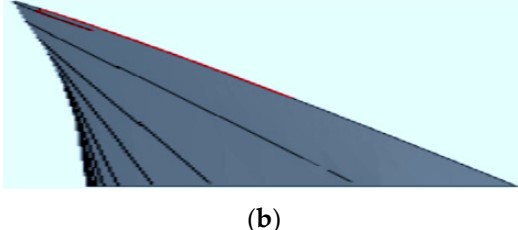
**(b)**

**Figure 24.** Crack growth changes of long and short half axes: (**a**) Crack growth variation of long half-axis; (**b**) Crack growth change of short half-axis.

### 5.2. Life Expectancy

The relationship between the number of cyclic loads and the size of the crack in the short half-axis is calculated and shown in Figure 25. According to IIW (International Welding Association) recommendations, when the structural material is steel, the values of fatigue crack growth parameters *C* and *m* in the Paris formula are as shown in Table 3.

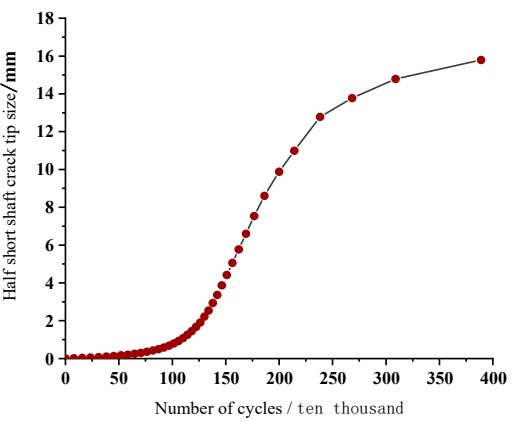

**Figure 25.** Relationship between cyclic load times and crack size of short half-axis.

**Table 3.** Parameter value of steel Paris formula.

| Kunit | C | m |
|---|---|---|
| N·mm$^{-3/2}$ | $5.21 \times 10^{-13}$ | 3 |
| MPa·m$^{1/2}$ | $1.65 \times 10^{-11}$ | 3 |

As can be seen in Figure 25, when the crack short half-axis size reaches 16 mm, the number of load cycles is nearly 3.9 million. That is, the fatigue life of the OSD model structure is about 3.9 million, while the number of fatigue cycle loads obtained from the test is nearly 3.5 million, with a relative error of 11.4%, which is within an acceptable range. The relative error between the model test and the FEM simulation results is inevitable as the effects of objective factors because crack closure effects are not taken into account in the simulation of crack expansion, and the expansion parameters $C$ and $m$ are estimated, whereas in practice, the values of $C$ and $m$ are different for different materials, different crack shapes and different stress ratios [40,41].

## 6. Influences of Different Parameters on Residual Life of Structure

The crack fatigue life of OSD may be affected by the initial defect size, the thickness and height of the U-rib, the thickness of the top plate and the thickness of the diaphragm [42,43]. Therefore, the method combining ABAQUS and Franc3D is used to analyze the influence of various parameters on the crack life, providing a reference for the future design of this kind of OSD structure.

### 6.1. Initial Crack Size

Because the OSD structure is very complex, coupled with environmental and manufacturing factors, it is inevitable to produce initial defects. Generally, the initial crack size can be obtained by a nondestructive evaluation method or an equivalent initial defect size method. In order to investigate the effect of the initial crack size on the remaining fatigue life of the OSD structure, the stress intensity factor amplitude $\Delta K_{\mathrm{I}}$ at the midpoint of the leading edge of the initial crack (tip of the short semi-axis) and the number of cyclic loads required to damage the structure (remaining fatigue life of the structure) are calculated for different sizes at a ratio of 3/2 between the long and short semi-axes. The relationship between the propagation depth of the middle point of the crack front and the amplitude of the stress intensity factor ($\Delta K_{\mathrm{I}}$) is shown in Figure 26, and the relationship between the propagation depth of the middle point of the crack front and the number of cyclic loads is shown in Figure 27.

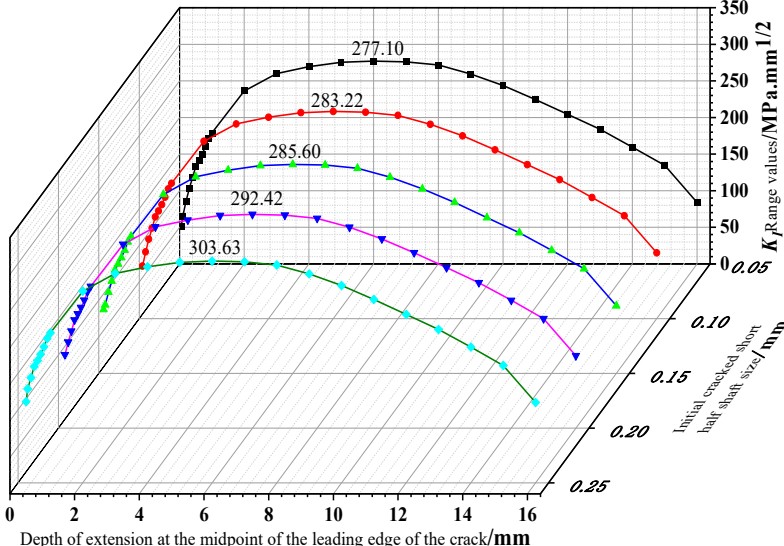

**Figure 26.** The relationship between the propagation depth at the middle point of crack front and the amplitude of the stress intensity factor.

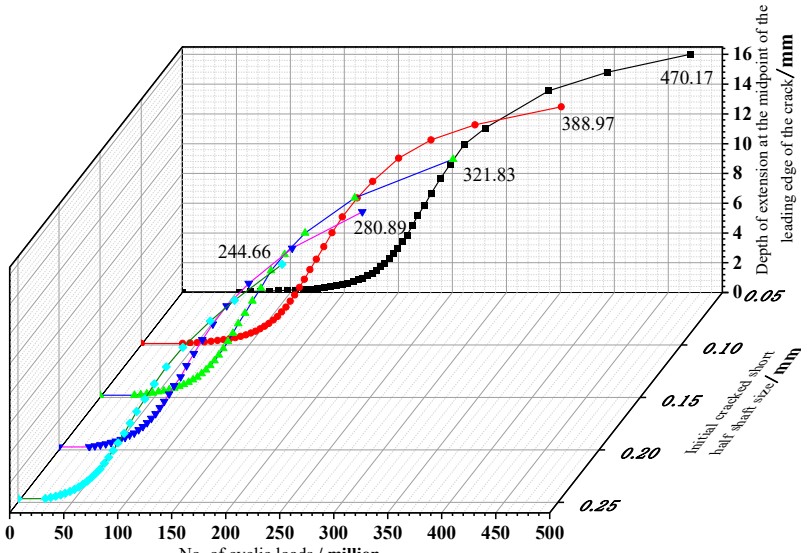

**Figure 27.** The relationship between the propagation depth at the middle point of the crack front and the number of cyclic loadings.

From the above two figures, it can be seen that the amplitudes of the stress intensity factor at the middle point of the crack front during the propagation of the initial cracks have the same shape, showing a trend of first increasing and then decreasing. The smaller the initial crack size is, the smaller the stress intensity factor amplitude at the crack front is, and the larger the residual fatigue life is.

### 6.2. Thickness of U-Rib

The different thicknesses of the U-rib and all other constant parameters are studied. The relationship between the propagation depth of the middle point of the crack front and the amplitude of the stress intensity factor ($\Delta K_I$) is shown in Figure 28, and the relationship between the propagation depth of the middle point of the crack front and the number of cyclic loads is shown in Figure 29.

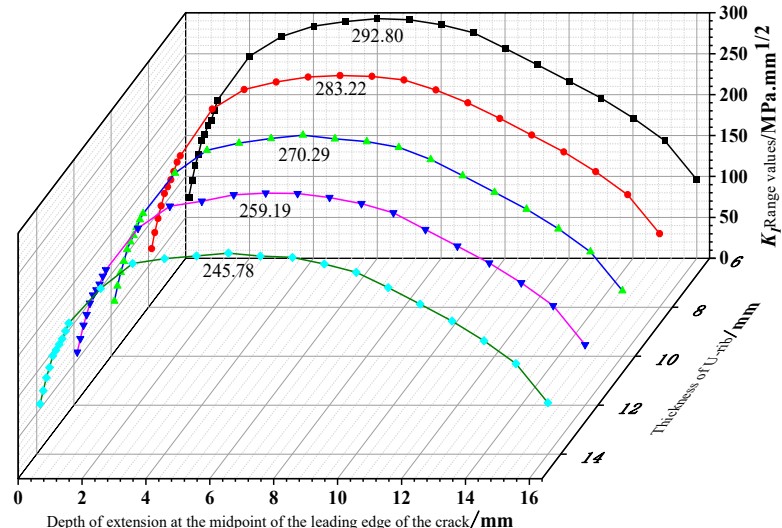

**Figure 28.** The relationship between the propagation depth at the middle point of the crack front and the amplitude of the stress intensity factor.

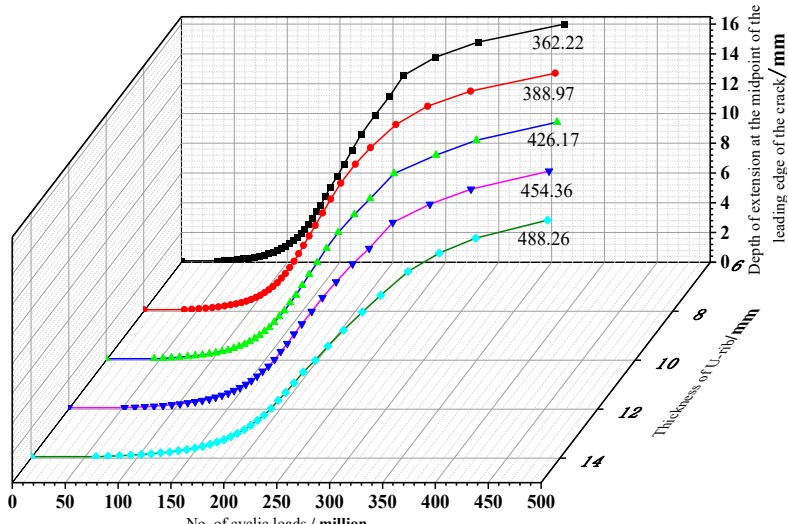

**Figure 29.** The relationship between the propagation depth at the middle point of the crack front and the number of cyclic loadings.

It can be seen from the above two figures that the thickness of the U-rib has a certain impact on the service life of the OSD structure. That is, with the increase in the thickness of U-rib, the amplitude of stress intensity factor at the middle point of the crack front decreases, and the fatigue life of the structure increases. This is probably due to the fact that the increase in U-rib thickness increases the stiffness of the structure as well as the load-carrying capacity; therefore, an appropriate increase in U-rib thickness can improve the service life of the structure while taking into account economic factors.

### 6.3. U-Rib Height

A simulation of the test model by the different heights of the U-rib and other constant parameters are performed [44]. The relationship between the propagation depth of the middle point of the crack front and the amplitude of the stress intensity factor ($\Delta K_I$) is shown in Figure 30, and the relationship between the propagation depth of the middle point of the crack front and the number of cyclic loads is shown in Figure 31.

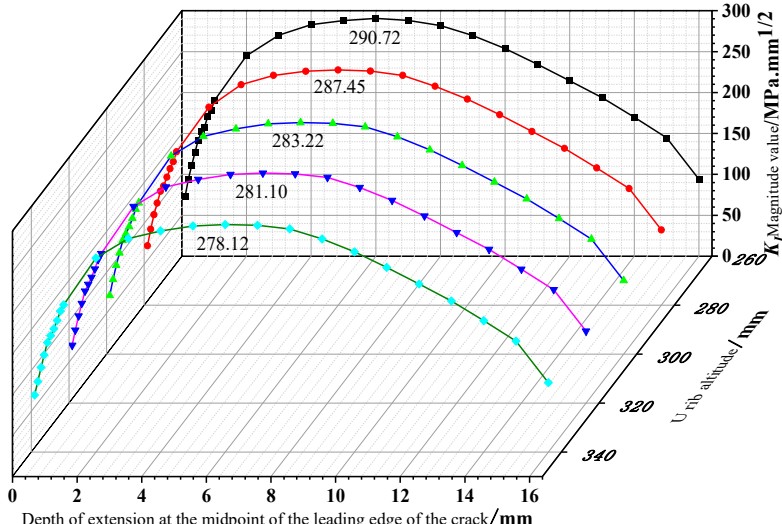

**Figure 30.** The relationship between the propagation depth at the middle point of the crack front and the amplitude of the stress intensity factor.

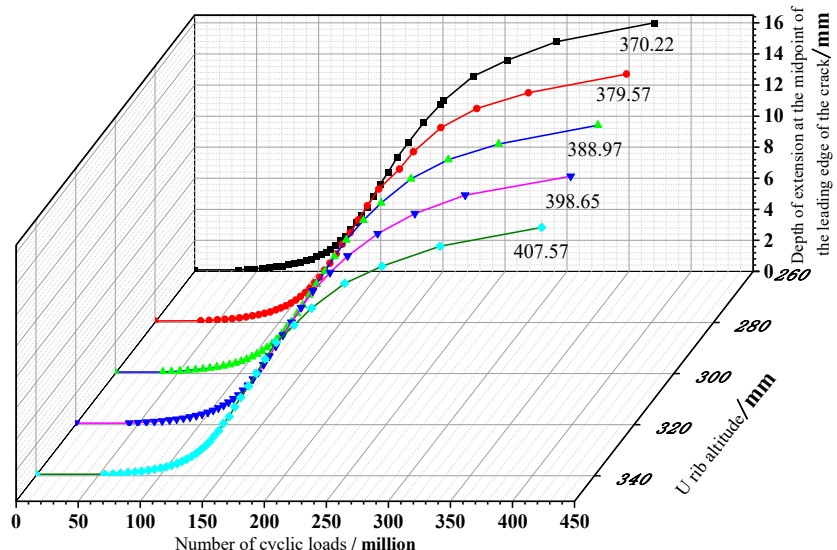

**Figure 31.** The relationship between the propagation depth at the middle point of the crack front and the number of cyclic loadings.

*6.4. Thickness of Top Plate*

The simulation of the test model by the different thicknesses of the top plate and all other constant parameters are conducted. The relationship between the propagation depth of the middle point of the crack front and the amplitude of the stress intensity factor ($\Delta K_{\mathrm{I}}$) is shown in Figure 32, and the relationship between the propagation depth of the middle point of the crack front and the number of cyclic loads is shown in Figure 33.

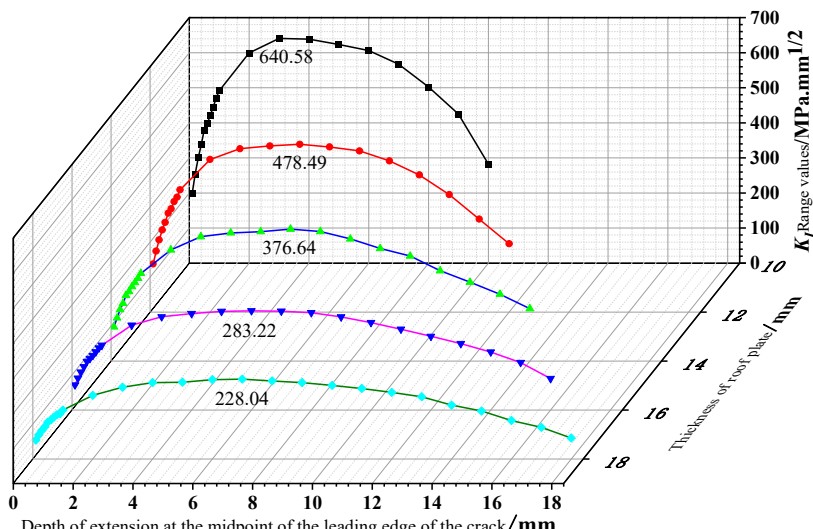

**Figure 32.** The relationship between the propagation depth at the middle point of the crack front and the amplitude of the stress intensity factor.

As can be seen from the above two figures, with the increase in the top plate thickness, the stress intensity factor amplitude at the middle point of the crack leading edge decreases significantly, and the stress intensity factor amplitude decreases by about 70% when the top plate thickness increases from 10 mm to 18 mm; with the increase in the top plate thickness the remaining life of the structure becomes longer and longer. An increase in plate thickness, while taking into account economic factors, will greatly help to improve the fatigue life of the structure.

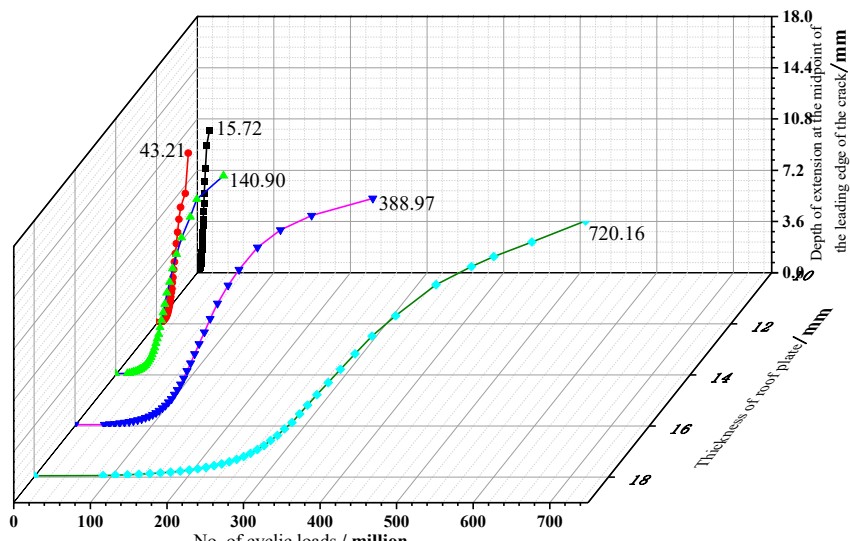

**Figure 33.** The relationship between the propagation depth at the middle point of the crack front and the number of cyclic loadings.

### 6.5. Thickness of the Horizontal Partition

The simulation analysis of the test structure is carried out by changing the diaphragm thickness without changing other parameters. The relationship between the propagation depth of the middle point of the crack front and the amplitude of the stress intensity factor ($\Delta K_{\mathrm{I}}$) is shown in Figure 34, and the relationship between the propagation depth of the middle point of the crack front and the number of cyclic loads is shown in Figure 35.

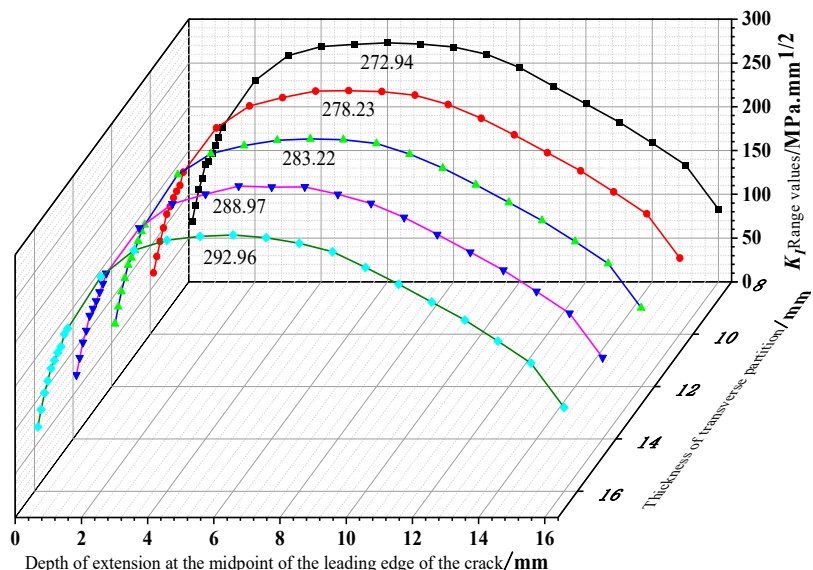

**Figure 34.** The relationship between the propagation depth at the middle point of the crack front and the amplitude of the stress intensity factor.

It can be seen from the above two figures that when the thickness of the diaphragm increases from 8 mm to 16 mm, the amplitude of the stress intensity factor at the middle point of the crack front does not decrease, but increases because the change amplitude is relatively small. Therefore, it can be seen that the thickness of the diaphragm has less impact on the service life of the structure.

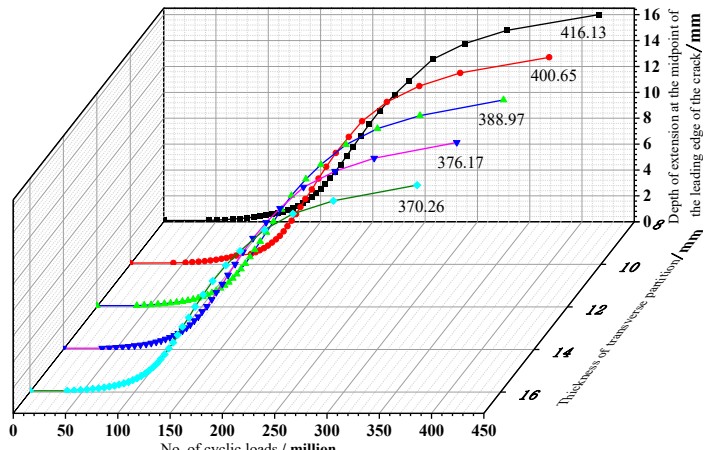

**Figure 35.** The relationship between the propagation depth at the middle point of the crack front and the number of cyclic loadings.

## 7. Conclusions

More and more steel bridges for urban rail transport are being built. However, because the vehicle load of a rail transit bridge is different from that of highways and railways, the stress characteristics and stress distribution of its OSD structure are also different. Therefore, the research on the fatigue of urban rail transit bridges has a certain degree of practical and theoretical significance.

In this paper, based on the OSD structure of an urban rail transit cable-stayed bridge, the fatigue test model is designed and carried out. The combination of ABAQUS and Franc3D is used to simulate the fatigue crack growth of the test, the law of the crack growth is studied, and the residual life of the structure is predicted according to the linear elastic fracture mechanics. The main conclusions are the following:

(1) A crack with a length of 15.1 cm appears near the boundary of the loading position for the first time after 3 million~3.25 million cycles of loading. After 3.25 million~ 3.5 million cycles of cyclic loading, the crack expanded to 18.6 cm in the model test;

(2) The finite element calculation results and the test results are basically the same, so the test can reflect the real state of the model force, and the test data have real reliability;

(3) The Mode I stress intensity factor of the initial crack is symmetrically distributed, and the value is large, reaching 77.67 MPa. The stress intensity factors of Type II and III fluctuate around 0 MPa·mm$^{1/2}$, which belongs to the crack type dominated by the Mode I crack;

(4) With the expansion of the crack, the stress intensity factor at the middle point of the leading edge of the crack tends to increase and then decrease. When the size of the short semi-axis of the crack reaches 16 mm, the number of load cycles is nearly 3.9 million, while the number of fatigue cycle loads tested is 3.5 million, with a relative error of 11.4%, which is within an acceptable range.

(5) The increases of U-rib thickness and roof thickness have the positive effect of prolonging the fatigue life of OSD. The influence of roof thickness is particularly significant. When the roof thickness increases from 10 mm to 18 mm, the amplitude of the stress intensity factor decreases by about 70%, which is more helpful in increasing the fatigue life.

**Author Contributions:** Conceptualization, Y.Z. and S.W.; methodology, Y.Z.; software, X.X. and S.W.; validation, X.X., S.W. and Y.Z.; formal analysis, X.X.; investigation, S.W.; resources, Y.Z.; data curation, H.T.; writing—original draft preparation, S.W.; writing—review and editing, H.T.; visualization, J.Z.; supervision, J.Z.; project administration, Y.Z.; funding acquisition, Y.Z. All authors have read and agreed to the published version of the manuscript.

**Funding:** This research was funded by [Natural Science Foundation of PR China, State Key Laboratory of Mountain Bridge, Tunnel Engineering Development Fund, and Chongqing Returned Overseas Students' Entrepreneurship and Innovation Support Fund] grant number [No.5227083397; CQSLBF−Y14, CQSLBF−Y16−10; cx2018113, cx2020117] And The APC was funded by [Natural Science Foundation of PR China].

**Institutional Review Board Statement:** This study does not involve human or animal research.

**Informed Consent Statement:** This study does not involve human research.

**Data Availability Statement:** Data available on request due to restrictions eg privacy or ethical. The data presented in this study are available on request from the corresponding author. The data are not publicly available due to [Due to the privacy related to the design of data content, it is not convenient to disclose all the data. The key data has been presented in the text of the article in a graphical table.

**Acknowledgments:** The authors appreciate the financial support from the Natural Science Foundation of PR China (Grant No.5227083397), State Key Laboratory of Mountain Bridge, Tunnel Engineering Development Fund (CQSLBF−Y14, CQSLBF−Y16−10), and Chongqing Returned Overseas Students' Entrepreneurship and Innovation Support Fund (cx2018113, cx2020117).

**Conflicts of Interest:** The authors declare no conflict of interest.

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
