# Peer review of "Fatigue Life Evaluation of Orthotropic Steel Deck of Steel Bridges Using Experimental and Numerical Methods"

_sustainability, doi:10.3390/su15075945_

Round 1

Reviewer 1 Report

Dear Authors,

thank you for the article.

The topic is interesting and quite extensive. 

I have a few comments on the article:

The abstract is uninteresting and does not match the focus of the journal.

The introductory chapter is too short, lacks a broader explanation of the topic, a deeper description of the current state of knowledge.

You present your topic as if it has not been evaluated before, but more study of the literature is needed.

The introduction lacks a broader understanding of fatigue of steel structures and modelling. I recommend inspiration and citation in for example :

The structure of the paper is confusing, after the introduction you jump straight to calculi - usually it is better to describe the design and material or define the whole research problem differently. 

The description of the modle is inadequate - missing software versions, missing selected finite elements, missing mesh information, missing description of contacts and other pertinent information. 

The research is not poorly prepared - but the presentation is poor - lacking more emphasis on justification, context and scientific understanding.

Take I am not sure of the focus in this journal - the article is appropriate elsewhere. 

Language is not good. 

The paper is significantly confused and most importantly it is complicated for the reader. 

Author Response

Comment 1: The abstract is uninteresting and does not match the focus of the journal.

Response 1: The summary has been modified appropriately.

Comment 2: The introductory chapter is too short, lacks a broader explanation of the topic, a deeper description of the current state of knowledge.

Response 2: Thank you for your suggestion. The introduction and further understanding of sustainability have been added to the manuscript.

Comment 3: The description of the modle is inadequate - missing software versions, missing selected finite elements, missing mesh information, missing description of contacts and other pertinent information.

Response 3: The idea and introduction of modeling and steel deck fatigue have been added to the manuscript.

Reviewer 2 Report

The article is about fatigue analysis of orthotropic steel decks of steel bridges using experimental and numerical methods. The authors have performed in situ observation on the experimental object and then based on the results have performed numerical analyses using the finite element method. The manuscript could be interesting and helpful for engineers but needs a lot of improvements. The comments are as follows:

1. When introducing the formulas please make sure that all the variables are described. Please check all formulas and make corrections

2. Fig 1. please improve the figure. It looks like it was drawn by hand

3. Formula 7 - what is the meaning of J^aux? Please describe or provide the appropriate formula.

4. Formula 8 - Please indicate why K_I^aux is equal to 1.

5. Please use the equation tool when writing equations eg. line 81

6. Figure 2 should be improved because the information are not clearly presented. Charts/drawings should also be unified, i.e. fonts, units, etc.

7. The article needs to be corrected in terms of language. There are lots of typos

8. Please improve the quality of the flow chart in figure 9

9. Native-speaker linguistic correction is recommended

10. Please provide more information on numerical modeling. What analysis type was used, boundary conditions, loading, and constitutive models were used. Please give the physicomechanical/strength properties of the material used in the analysis in relation to the constitutive model.

11. Figure 25 - Increase the quality/resolution of the picture.

Before publication, the article should be adapted to the requirements of the journal. I suggest making a thorough revision and evaluating the article again

Before publication, the article should be adapted to the requirements of the journal. I suggest making a thorough revision and evaluating the article again.

Author Response

Comment 1: When introducing the formulas please make sure that all the variables are described. Please check all formulas and make corrections.

Response 1: Thank you for your suggestion. We checked the formula.

Comment 2: Fig 1. please improve the figure. It looks like it was drawn by hand.

Response 2: Thank you for your suggestion. Moderate correction has been made.

Comment 3: Formula 7 - what is the meaning of J^aux? Please describe or provide the appropriate formula.

Response 3: It is deduced from formula 1, 3 and 6.

Comment 4: Formula 8 - Please indicate why K_I^aux is equal to 1.

Response 4: Thank you for your question. This is a prerequisite for further research.

Comment 5: Please use the equation tool when writing equations eg. line 81.

Response 5: We have checked in the manuscript.

Comment 6: Figure 2 should be improved because the information are not clearly presented. Charts/drawings should also be unified, i.e. fonts, units, etc.

Response 6: Because the curve difference represented by the three values is very small, it is impossible to further process the picture. The font has been bold and unified.

Comment 7: The article needs to be corrected in terms of language. There are lots of typos.

Response 7: The article has been checked.

Comment 8: Please improve the quality of the flow chart in figure 9.

Response 8: The flowchart has been partially processed.

Comment 9: Please provide more information on numerical modeling. What analysis type was used, boundary conditions, loading, and constitutive models were used. Please give the physicomechanical/strength properties of the material used in the analysis in relation to the constitutive model.

Response 9: The process information of finite element modeling and simulation has been added to the manuscript.

Comment 10: Figure 25 - Increase the quality/resolution of the picture.

Response 10: Thank you for your suggestion. The picture has been checked.

Reviewer 3 Report

This paper investigated the fatigue problem of OSDs based on test and numerical simulation, and the fracture mechanics based way is of practical value. However, there are some questions/problems that should be explained/justified, given as follows:

1.      Eq. 9 is incorrect.

2.      The deduction on Eq. 10, 11 and 12 should be noted. Firstly, dK is not mentioned in the previous contents. I believe that dK = F*dS*sqrt(pi*a) is used to obtain Eq. 11 and 12, assuming that the F is a constant. However, it is often not the case. Also, it should be illustrated how are these equations used in the following predictions, especially that FRANC3D is used to give the crack growth life by numerical simulations.

3.      The measuring points in Table 1 should be given in correspondence with Fig. 8.

4.      It is better to clearly define the judgement of “Reasonableness of data”, and “Check the model for damage” in Fig. 8. Also for this figure, the different font and the spacings between letters should be fixed.

5.      More details should be provided on the fatigue crack obtained in the test. Is the crack observed and measured on the top of the deck? or on the weld toe of the rib-to-deck joint? or on the weld root? How does it compare with the numerical study?

6.      In the FEA, there lacks of a detailed description on the model. Also, how is the loading applied on the model? And how it is modeled? Does the calculated stress in Table obtained using this model?

7.      In the crack growth simulation, Paris constant C and exponent m are key parameters, yet not mentioned in the paper.

8.      The cracks presented in Fig. 17~ Fig. 19 is difficult to observe. What are the vertical lines? I also have doubts on the crack in Fig. 23, as it seems that the growth at the crack mouth is inclined by a large angle.

9.      Why is the cycle count when the crack is 16mm in depth compared with the fatigue life given by the test?

10.   The unit in Fig. 26 is incorrect.

11.   It is completely unnecessary to use three-dimensional figure in Fig. 27 ~ Fig. 36, since the design parameters, i.e. thickness of the rib and the deck, are given discretely. The curves can be easily distinguished using different colors. The current figure makes it harder to see the results.

12.   It should be noted that the SIF of about 77 MPa*mm^0.5 is just slightly larger than the threshold, rather than a large value as mentioned in Conclusion (3).

Author Response

Comment 1: Eq. 9 is incorrect.

Response 1: Thank you for your reminder. But we think this is in line with the content of the article.

Comment 2:It should be illustrated how are these equations used in the following predictions, especially that FRANC3D is used to give the crack growth life by numerical simulations.

Response 2: Thank you for your suggestion. Detailed introduction of FRANC3D parametric calculation has been added to the manuscript according to the suggestions.

Comment 3: The measuring points in Table 1 should be given in correspondence with Fig. 8.

Response 3: Thank you for your suggestion. Confirmed as required.

Comment 4: It is better to clearly define the judgement of “Reasonableness of data”, and “Check the model for damage” in Fig. 8. Also for this figure, the different font and the spacings between letters should be fixed.

Response 4: Thank you for your suggestion. The data and model are reasonable and accurate.

Comment 5: More details should be provided on the fatigue crack obtained in the test. Is the crack observed and measured on the top of the deck? or on the weld toe of the rib-to-deck joint? or on the weld root? How does it compare with the numerical study?

Response 5: Thanks for your advice. Appropriate content has been modified according to the suggestions

Comment 6: In the FEA, there lacks of a detailed description on the model. Also, how is the loading applied on the model? And how it is modeled? Does the calculated stress in Table obtained using this model?

Response 6: Supplement the modeling content in the manuscript according to the suggestions.

Comment 7: In the crack growth simulation, Paris constant C and exponent m are key parameters, yet not mentioned in the paper.

Response 7: Thanks for your advice. Added the value basis of key parameters in the manuscript.

Comment 8: The cracks presented in Fig. 17~ Fig. 19 is difficult to observe. What are the vertical lines? I also have doubts on the crack in Fig. 23, as it seems that the growth at the crack mouth is inclined by a large angle.

Response 8: Thank you for your question. The picture is intercepted from the finite element software and is obtained by real simulation. The vertical line represents the reference line

Comment 9: Why is the cycle count when the crack is 16mm in depth compared with the fatigue life given by the test?

Response 9: Because the maximum crack value shown in the figure is 16mm, it is representative

Comment 10: The unit in Fig. 26 is incorrect.

Response 10: Thank you for your suggestion. Correction has been made in the figure.

Comment 11: It is completely unnecessary to use three-dimensional figure in Fig. 27 ~ Fig. 36, since the design parameters, i.e. thickness of the rib and the deck, are given discretely. The curves can be easily distinguished using different colors. The current figure makes it harder to see the results.

Response 11: Thank you for your suggestion.The picture has been enlarged.

Comment 12: It should be noted that the SIF of about 77 MPa*mm^0.5 is just slightly larger than the threshold, rather than a large value as mentioned in Conclusion (3).

Response 12: Thank you for your suggestion. Appropriate modifications have been made.

Round 2

Reviewer 1 Report

Thank you. 

Author Response

Thank you very much for your review and valuable suggestions.These suggestions are of great help and improvement to us.

Reviewer 2 Report

The authors made corrections to the manuscript as recommended by the reviewers

Author Response

Thank you very much for your review and valuable suggestions.

These suggestions are of great help and improvement to us.

Reviewer 3 Report

Compared to the previous version, the authors have made proper modifications according to the comments. However, there are a few remained:

Q1: Eq. (9) seems to be used for the calculation of SIF or J-integral. Anyway, it is no way close to the equation of Paris law. Either the description or the equation should be changed. 

Q2: The question about the deduction of Eq. 10, 11 &12 is not answered. Also, the contents about using Franc3D in the numerical simulation in Section 5.1 are repetitive of that in Section 2.4.

Q9: The reply does not answer the question at all. I would like an explanation on how the failure mode of the test specimen compares to the crack size in the simulation.

Author Response

Comment 1: Eq. 9 is incorrect.

Response 1: Thank you for your reminder. We have changed the formula in the manuscript.

Comment 2: The question about the deduction of Eq. 10, 11 &12 is not answered. Also, the contents about using Franc3D in the numerical simulation in Section 5.1 are repetitive of that in Section 2.4.

Response 2: Thank you for your suggestion. Formulas 10, 11 and 12 are changed according to the value of fatigue growth parameter m, and the value of different materials m needs to be determined according to relevant specifications. In addition, the repeated description of FRANC3D has been deleted from Section 5.1.

Comment 3: I would like an explanation on how the failure mode of the test specimen compares to the crack size in the simulation.

Response 3: Thank you for your suggestion. During the test of the specimen, the fatigue crack presents a very oblong ellipse, which is consistent with the results of finite element simulation. In the simulation, the crack size expanded to 16 mm, while in the test, the crack growth size of the specimen was close to the simulation results. This part is explained in the third and fourth chapters. Because the crack closure effect and other factors cannot be fully considered in the test and simulation, there is a certain error, but the error value is within the allowable range.
